# Additive Manufacture of 3D Auxetic Structures by Laser Powder Bed Fusion—Design Influence on Manufacturing Accuracy and Mechanical Properties

**Sibi Maran, Iain G. Masters and Gregory J. Gibbons *** 

Additive Manufacturing Group, WMG, The University of Warwick, Coventry CV4 7AL, UK;
sibimaranpg@gmail.com (S.M.); I.G.Masters@warwick.ac.uk (I.G.M.)
* Correspondence: g.j.gibbons@warwick.ac.uk; Tel.: +4424-7652-2524

**Featured Application: A potential application of this work is in the manufacture of energy absorbing, light-weight structures such as crash protection safety wear.**

**Abstract:** The mechanical response of steel auxetic structures manufactured using laser-powder bed fusion was explored. The level of control exerted by the key design parameters of vertical strut length (H), re-entrant strut length (L), strut thickness (t) and re-entrant angle ($\theta$) on the mechanical response was examined through a design of experiment approach with ANOVA statistical analysis methods applied. The elastic modulus in directions normal to ($E_x$) and parallel to ($E_y$) the vertical strut was found to be primarily dependent upon t and L, respectively, whereas yield strength in both test directions ($\sigma_x$ and $\sigma_y$) was strongly dependent on t and L. A large variation in modulus was found between the two test directions ($E_x$ / $E_y$ − 1.02 ± 0.07 GPa/ 4.4 ± 0.1 GPa), whereas, yield strength showed little anisotropy ($\sigma_x$ / $\sigma_y$–45 ± 6 MPa/ 45 ± 9 MPa). Poisson's ratio parallel to the vertical strut varied considerably with geometry but not in a direction normal to the vertical strut. Deformation mechanisms were found to be different of compression in the x and y directions, being a combination of stretching of the vertical strut; compression, bending and hinging of the re-entrant strut (x); and vertical strut compression and re-entrant strut stretching and bending (y).

**Keywords:** additive manufacture; auxetic; mechanical; manufacturing; accuracy

## 1. Introduction

Material with negative Poisson's ratio was initially observed by Love in 1892 in the form of single crystal pyrite with Poisson's ratio -0.14 [1]. The term "auxetic" was coined in 1991 by Evans [2] for materials with negative Poisson's ratio, which is derived from the Greek word "auxetikos" which means "that which tends to increase" [2]. Re-entrant geometry is formed by inverting the diagonal struts in the normal hexagonal honeycomb geometry. Re-entrant geometry was initially presented by Gibson [3], and then by Almgren [4] who developed auxetic structures made of rods, hinges and springs based on re-entrant geometry. Wojciechowski developed computer models of hard [5] and soft [6] cyclic hexamers and demonstrated that they exhibited phases having negative Poisson's ratio, and more recently, a strongly anisotropic chiral phase of negative Poisson's ratio was found in a work about hard cyclic tetramers [7]. Re-entrant structure was also observed in auxetic foams formed by the volumetric compression of conventional foams, with ribs that permanently protrude inwards forming a re-entrant structure [8]. Re-entrant honeycombs also deform by flexure, stretching of the struts, as well as the hinging mechanism of the strut joints [9]. A wide range of other auxetic structure models has been developed, including the triangular variation of re-entrant structure, proposed by

Larsen et al [10]; the chiral honeycomb model, first introduced by Lakes [11], and further described by Prall and Lakes [12], made up of circular nodes connected by the straight ligaments in a tangential manner; the rotating rigid unit model, which was first introduced by Grima and Evans [13] and, independently, by Ishibashi and Iwata [14] in the same year, in the form of rigid squares connected at the vertices by hinges; the missing rib model, proposed by Smith [15] as an alternative to the re-entrant model due to presence of the broken ribs in transformed auxetic foams; and the nodule fibril model, proposed by Alderson and Evans [9], consisting of a network of nodules interconnected by fibrils.

Auxetic structures are known for their enhanced mechanical, electromagnetic and acoustic properties. Mechanical properties include increased indentation resistance, energy dissipation capacity, bending stiffness, toughness, shear resistance and fatigue resistance [16]. Apart from mechanical properties, auxetic structures also possess increased di-electric and sound absorption capability compared to conventional materials [17,18]. Due to their enhanced properties, auxetic structures find application in various fields, namely, automotive, aerospace, military, biomedical etc. Some of the applications include sandwich structures, energy absorbing materials, bio-medical implants, knee cap, blast proof curtains, safety jackets, helmets etc. [16,19,20].

There has been a recent rapid growth in interest in auxetic structures, particularly in their application in civil engineering, e.g., as damping elements for seismic protection [21], and as damage tolerant building materials [22], as well as new smart metamaterials [23,24].

Though various 2D auxetic models have been previously manufactured successfully, it has been a challenge to manufacture 3D auxetic structures, especially in metal. Due to their structural complexity, they cannot be manufactured using conventional manufacturing processes. Initially, Choi and Lakes [25] transformed conventional metallic foams into auxetic foams with 3D re-entrant structures using a thermal compression and expansion process, although this provided limited control over the dimensional and mechanical properties of the structure.

Powder bed fusion (PBF) additive manufacturing (AM), a layer AM process, building parts via the selective laser fusion of metal powder layers, offers the ability to manufacture complex metal parts, with lattice structure [26,27] and, therefore, could enable manufacture of complex 3D auxetic structures. McKown [28] manufactured different unit cell combinations of octahedral and pillar micro-lattice structures using PBF to be used for load bearing applications. The cross-section of the fabricated struts exhibited elliptical shape as opposed to the designed circular shape, impacting the mechanical behaviour of the samples. Rehme [27] provided a comprehensive summary of cellular structures and in-depth analysis and validation of PBF capabilities, providing insight into the effect of processing parameters on strut quality and concluding that good quality struts are produced by minimum exposure time and minimum exposure area, and indicating a strong relationship between strut diameter and amount of energy deposited. Hussein [29] manufactured four different cell types of triply periodic minimal surface (TPMS) cellular structures from three metallic powders 316L, Ti-6Al-4V and AlSi10 Mg) using PBF. Computer tomography (CT) analysis carried out on the samples showed no major geometry defects, which confirmed the ability of PBF to manufacture a wide range of cellular structures. An increase in the thickness of the struts was observed due to the increased energy concentration on the small struts. Hasan [30] performed detailed analysis of mechanical, dimensional and geometrical properties of Ti-6Al-4V body-centred cubic (BCC) micro-lattice structures manufactured by PBF, and demonstrated the benefit of a heat treatment in improving the mechanical properties and dimensional accuracy of the samples. Contuzzi et al. [31] investigated scanning strategies for PBF of micro-lattices and concluded that random paths with track overlapping as a strategy to improve density and mechanical properties. Li studied the possibility of manufacturing 3D re-entrant structures using PBF from shape memory alloys (SMA) [32], however, Li was forced to modify the auxetic design to overcome the limitation of PBF in manufacturing overhanging structures, which compromised Poisson's ratio. Schwerdtfeger [33] and Yang [34] manufactured metal 3D auxetic structures with re-entrant configuration using the selective electron beam melting (SEBM) process, but the fabricated structures had issues with quality in the form of surface roughness and dimensional

accuracy, which prevented the structures being used for bio-medical applications, especially in implants such as hip stems and spinal vertebral discs [35].

There is, therefore, a need for a manufacturing process that can provide good control over surface roughness, dimensional accuracy and mechanical properties of the auxetic structures. There has been no systematic study of the capability of PBF in manufacturing a wide range of re-entrant unit cell configurations and the influence of unit cell design parameters on their mechanical properties. There are also contradictions between the experimental mechanical test results of PBF re-entrant structures and theoretical models due to the induced torsional moments, internal stresses in the struts, and manufacturing defects [34]; thus, there is need to verify the assumptions made for the deforming mechanism in Yang's theoretical model using experimental methods such as digital image correlation (DIC).

This research therefore aims to address these gaps by studying the feasibility of using laser-PBF for a wide range of re-entrant unit cell configurations and developing a methodology which forms a basis for using PBF for manufacturing metallic 3D auxetic structures.

## 2. Materials and Methods

This research focuses on studying the manufacturability and mechanical properties of 3D re-entrant auxetic structures. Therefore, 24 half-factorial design of experiments (DoE) were designed, as shown in Table 1, to analyse the manufacturability of different design configurations by PBF and determine the influence of design parameters on their mechanical properties. Four unit cell design parameters, namely, length of vertical strut "H", length of re-entrant strut "L", re-entrant angle "θ" and strut thickness "t" were taken into consideration, as shown in Figure 1. The values selected for H, L and θ were based on an initial virtual analysis of the buildability of the structures within the CAD environment. The values selected would, theoretically, enable all samples to be manufactured successfully (without failure and with no use of supporting structures), subsequently allowing subsequent ANOVA to be performed.

**Table 1.** Design of experiment for auxetic structure design.

| Sample | H (mm) | L (mm) | θ (○) | t (mm) |
|--------|--------|--------|-------|--------|
| 1 | 6.5 | 2.5 | 45 | 0.8 |
| 2 | 7 | 2.5 | 45 | 1 |
| 3 | 6.5 | 3.5 | 45 | 1 |
| 4 | 7 | 3.5 | 45 | 0.8 |
| 5 | 6.5 | 2.5 | 70 | 1 |
| 6 | 7 | 2.5 | 70 | 0.8 |
| 7 | 6.5 | 3.5 | 70 | 0.8 |
| 8 | 7 | 3.5 | 70 | 1 |

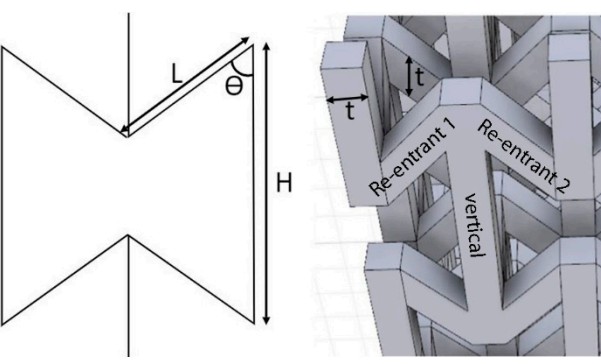

**Figure 1.** Re-entrant unit cell parameters.

From the research carried out by Yang el al., it could be seen that there is no variation in the mechanical properties once the repetition of the unit cell is greater than 3, and the structures behave as a continuum [36]. Hence, samples with $4 \times 4 \times 4$ unit cells were designed for manufacture. The eight corresponding structures are given in Figure 2. The structures were designed using SolidWorks CAD software (Dassault Systèmes, Coventry, UK).

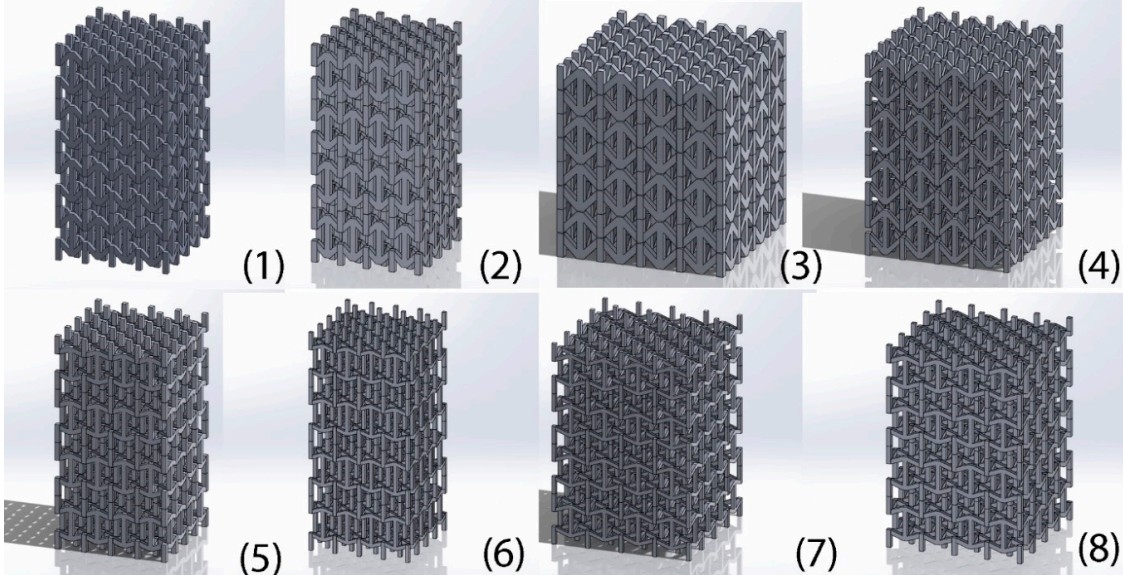

**Figure 2.** CAD models of the 8 design variations. (1–8) refer to samples 1–8. The values selected for H, L and θ were based on an initial virtual analysis of the buildability of the structures within the CAD environment. The values selected would, theoretically, enable all samples to be manufactured successfully (without failure and with no use of supporting structures), subsequently allowing subsequent ANOVA to be performed.

All samples were manufactured in maraging steel (MS1) from EOS GmbH (EOS GmbH, Eschweiler, Germany) using an EOS M280 PBF system, with a 400W laser at 40μm layers under a nitrogen atmosphere. The CAD designs were orientated at 35° to the XY (horizontal build plane) so as to exclude the use of internal support structures, requiring only external support structures. A solid base support structure was added to the designs to act as a heat-sink to prevent over-heating of the structures. An example part set-up is shown in Figure 3. It was necessary to angle sample 8 at 10° to the XY plane in order to compensate for the angle of build of the long struts. This is below the accepted angle for this material for an unsupported surface (35°), however, as the length of the re-entrant strut is short, the machine is capable of building the strut at the shallow 10° angle without failure. Default build parameters for MS1 [37] were modified to reduce energy levels to prevent over-heating. Non-standard laser power, scanning speed and hatch spacing was used for up skin, in skin and down skin (Figure 4a), using the parameters given in Table 2. Stripes scanning configuration was used, as shown in Figure 4b.

**Table 2.** Scan parameters employed for sample manufacture.

| Layer | Laser Power (W) | Scan Speed (mm/s) | Hatch Spacing (μm) |
| --- | --- | --- | --- |
| Up Skin | 165 | 600 | 110 |
| In Skin | 305 | 1010 | 90 |
| Down Skin | 150 | 2000 | 60 |

Four sets of each of the 8 designs (Table 1) were manufactured. An example build of one set of the 8 designs is given in Figure 5. After the PBF process, the samples were removed from the base plate via wire Electro Discharge Machining (EDM) using a V650G (Excetek Technologies Co. Ltd., Taichung City, Taiwan).

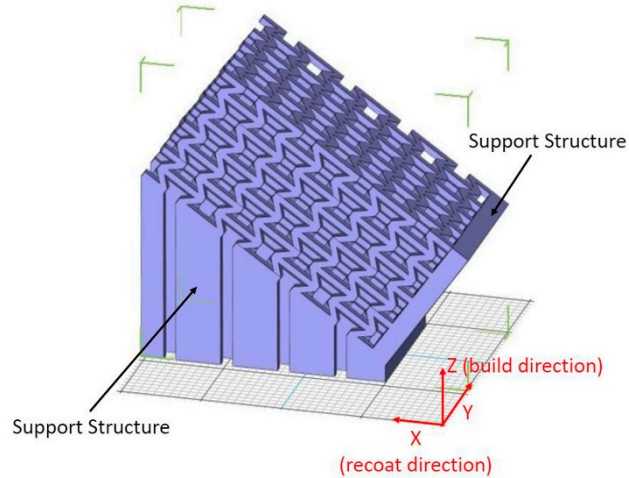

**Figure 3.** Illustration of sample alignment for manufacture.

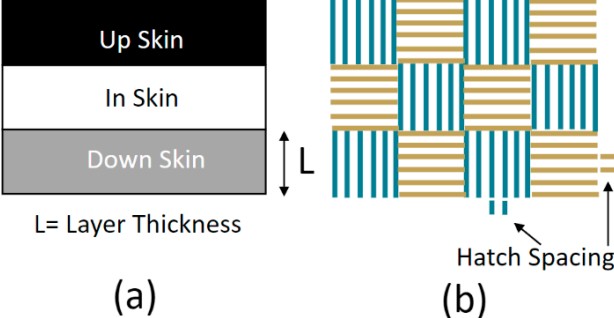

**Figure 4.** Illustration of (**a**) the scan layer nomenclature used and (**b**) laser scan pattern used.

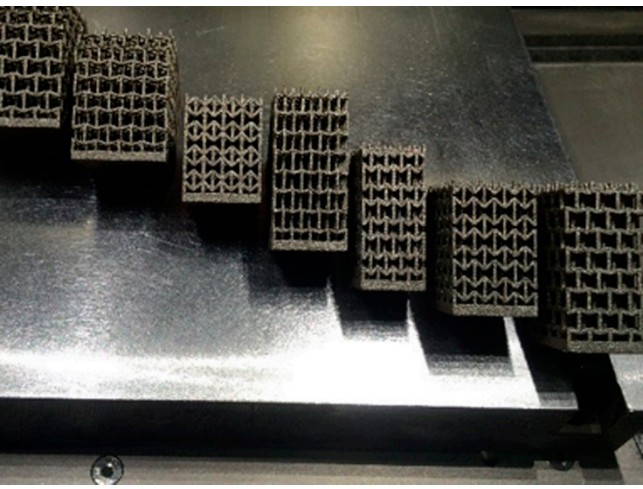

**Figure 5.** Example of each of the 8 designs fabricated using powder bed fusion (PBF).

Sample overall dimensions were measured using a vernier calliper, and their mass was measured using a digital balance. From those measurements, the density was calculated for each sample. The relative density was calculated using the known MS1 bulk density of 8.05 ± 0.05 gcm$^{-3}$ [37].

The theoretical density of the structures was also calculated from the CAD data in SolidWorks CAD software (Dassault Systèmes, Coventry, UK), using the CAD volume for each structure for comparison.

Sample 3 and 4 were selected for metallurgical analysis as these samples contain a 45° re-entrant angle, which poses the largest risk for build failure and strut deformation, as the oriented struts in the unit cell are at the shallowest angle with respect to build orientation (compared to samples with a 70° re-entrant angle). The samples were cut along the cross-sectional and longitudinal axes using an AbrasiMet 250 manual abrasive cutter (Buehler UK, Coventry, UK). The samples were hot mounted in EpoMet™ G (Buehler UK, Coventry, UK). The samples were then ground and polished using standard metallurgical procedure using an Eco Met 250 Pro (Buehler UK, Coventry, UK). The samples were finished with colloidal silica polish (MasterPolish 0.06μm, (Buehler UK, Coventry, UK)).

The strut thickness and porosity of the sample struts were analysed using optical microscopy, using an Eclipse MA200 inverted materials optical microscope (Nikon Metrology, Castle Donnington, UK) at 5–100x magnification. Samples were images as manufactured with no post-processing or surface finishing. The measure of porosity was taken at the centre of the cross-sectional area of each strut, taken in the XZ plane (see Figure 3), as carried out by Casalino [38]. As the edges are not uniform, it is difficult to specify the measurement area if the whole cross-section is taken for examination. The internal porosity was measured for five vertical, five re-entrant 1 and 5 re-entrant struts for each sample. The porosity and circularity of the pores were extracted from the images using Image J2 software. Strut thickness was measured using image analysis of the optical microscope images using Image J2 software (ImageJ, Open source software) with 20 measurements taken across each strut to obtain average values.

Compression testing was performed using a 5800R tensile test system with 100kN load cell (Instron, High Wycombe, UK). Two smooth steel plates were used on either side of the auxetic part in order to reduce friction and to prevent any damage to the fixtures. Testing was performed at 25 °C using a cross-head speed of 1 mm/min. An ARAMIS non-contact measuring system (GOM UK Ltd, Coventry, UK) was used to capture the deformation of samples during compression using the digital image correlation (DIC) technique. The DIC data were also used to measure strut lengths and re-entrant angles; 5 measurements were taken per feature measurement. The test directions, X and Y, are shown in Figure 6.

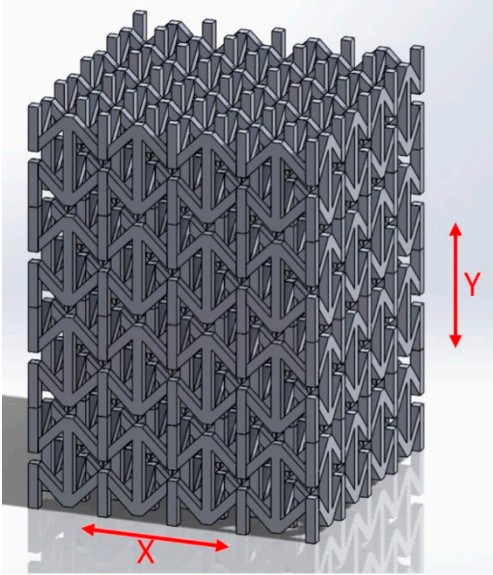

**Figure 6.** Uniaxial compression test directions.

## 3. Results

### 3.1. Density and Dimensional Accuracy

The results of the density measurement of the auxetic structures and the CAD-derived values are given in Figure 7. Errors in fabricated parts and Δ are ± 1%.

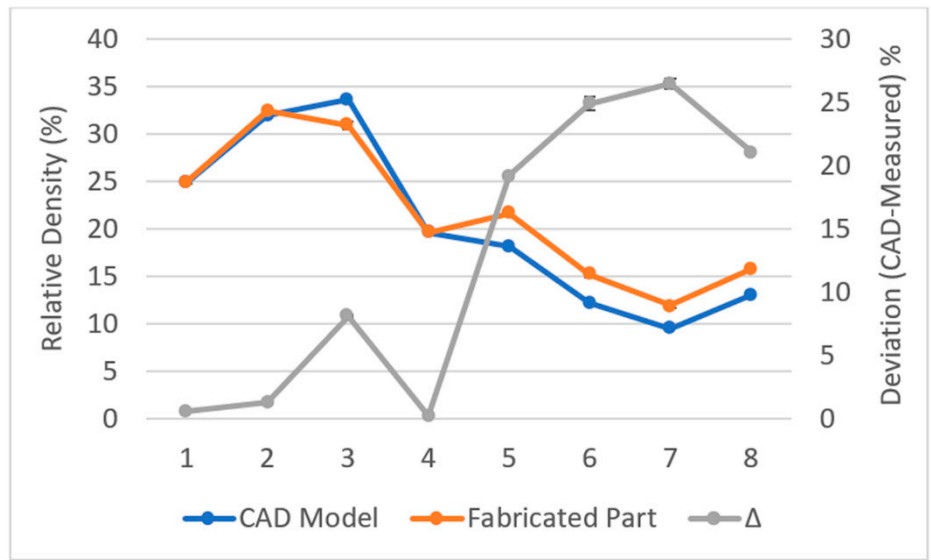

**Figure 7.** Theoretical CAD relative density and measured relative density of the auxetic structures.

From Figure 7, it can be seen that the relative density of most of the fabricated samples is greater than that calculated from the CAD models. The increase in relative density is likely due to the increase in thickness of the struts of the fabricated samples. The strut size deviations from CAD for the three struts in Samples 3 and 4 are given in Figure 8. It can be seen that for both samples, strut re-entrant 2, built at an angle of 80° to the build-platform, is the most accurate strut and closer to the designed thickness of 1000 μm. Furthermore, as expected, re-entrant 1, which was built almost horizontally at an angle of just 10° to the build-platform without any support structures, has the worst dimensional accuracy, with increased thickness over the designed value, and therefore, the thickness of the strut increases as the build angle of the struts decreases. All struts, other than the re-entrant 2 struts, are thicker than the design intent, and thus account for the disparity between the CAD and actual values for apparent density in Figure 7. The fabricated samples also show a wide range of variation in relative density (0.565 ± 0.005% to 26.5 ± 0.4%). Those built with a re-entrant angle of 70° show significantly higher deviation (average of 3 ± 2%) than those at 45° (average of 23 ± 2%). The use of a higher re-entrant angle would align re-entrant strut 1 to a greater angle to the build-plate (55°) without compromising greatly the orientation of re-entrant strut 2 (105°), which in in agreement with the strut thickness trend observed for samples 3 and 4.

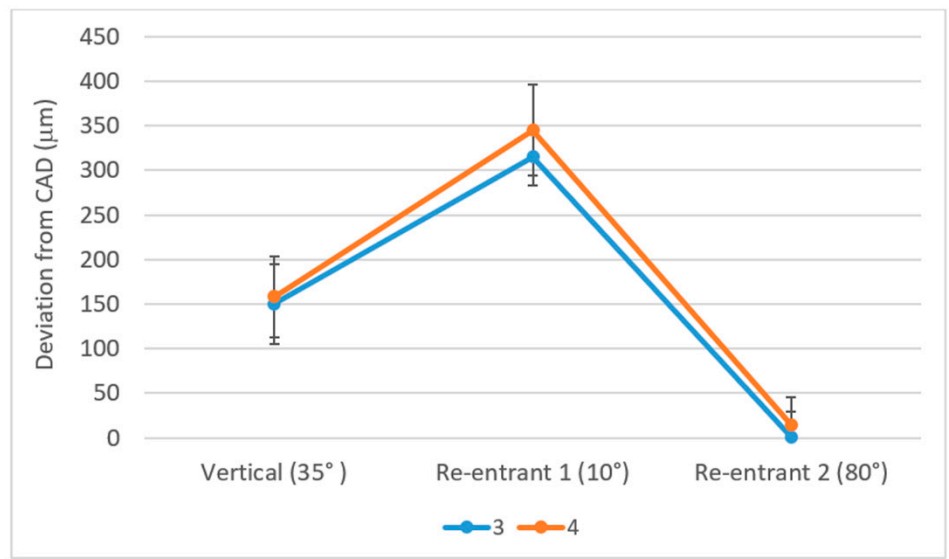

**Figure 8.** Strut size deviation from CAD for samples 3 and 4.

Strut lengths were obtained from the DIC image data. An example of a DIC image used is given in Figure 9. The deviations from the nominal CAD sizes (Table 1) are given in Figure 10.

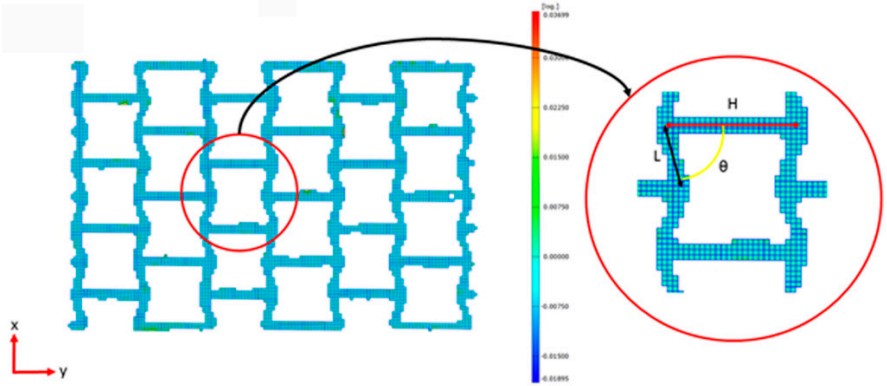

**Figure 9.** Digital image correlation (DIC) image used to measure unit cell dimensions.

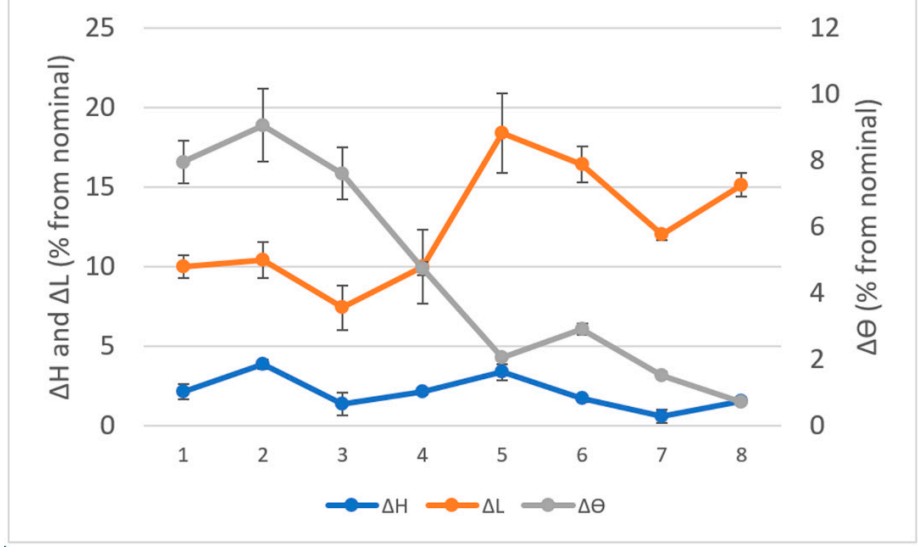

**Figure 10.** Unit cell dimension deviations from nominal.

There was a small increase in the length of the vertical struts (ΔH) by an average of 2.1 ± 0.4%. For the re-entrant struts, there was an average deviation of 12 ± 1%, which may be a result of their shallower build angle with respect to the build-plate (XY-plane) (see Figure 3 for orientation reference). An average of 5 ± 1% deviation in re-entrant angle was observed across all structures, but was considerably less for the 70° structures than for those with a 45° angle. Again, this is likely due to the increasing inaccuracy in geometry for features orientated at lower angle to the XY plane. The results of the porosity analysis are given in Figure 11.

The relative density for all the struts is greater than 99%. The re-entrant 2 struts, which are inclined at 80° to the build-plate, proved to be the best orientation, with a relative density as high as 99.9%. This is similar to the findings of other work which showed that vertically oriented struts/samples provide maximum density [39]. Wauthle also observed that struts oriented horizontally were the most affected in terms of porosity, followed by diagonal struts [39], however, in this work, the vertical strut (inclined at 35°) to the build direction (almost diagonally oriented) expressed slightly higher porosity when compared with re-entrant 1, which was inclined at 10° (almost horizontally oriented during build). The circularity results indicate that the pores present in all struts are regularly shaped and circular (value approximately 1), with the circularity of the pores increasing slightly as the build angle of the struts increases.

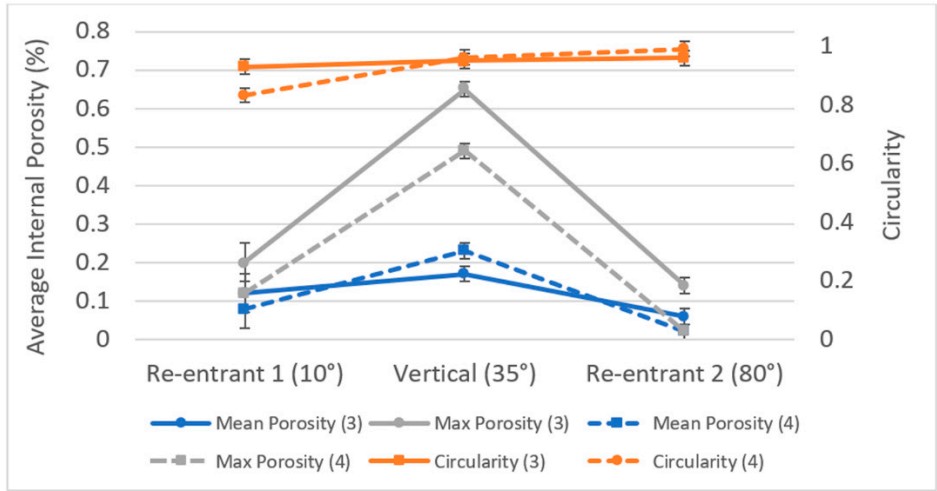

**Figure 11.** Internal porosity and pore circularity for samples 3 and 4.

When the whole cross-section of a strut is analysed, it can be seen that the porosity concentration is higher in the outer area, and the pores become larger and irregularly shaped (Figure 12). Dross formation occurs in horizontally built parts on down-facing unsupported layers (down skins) due to molten metal sinking under gravity. This leads to increased dimension and surface roughness of the parts [40–42], however, the dross formation can be minimised in the horizontally built parts by reducing the energy input by reducing the permeation of the laser into the powder below the layer [42], [29]. The formation of dross is seen in the down skin struts of the lattices (Figure 13).

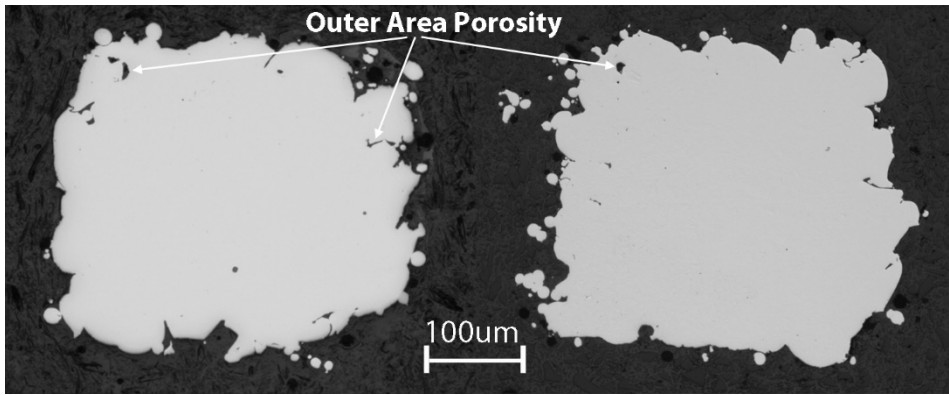

**Figure 12.** Outer-edge porosity of struts.

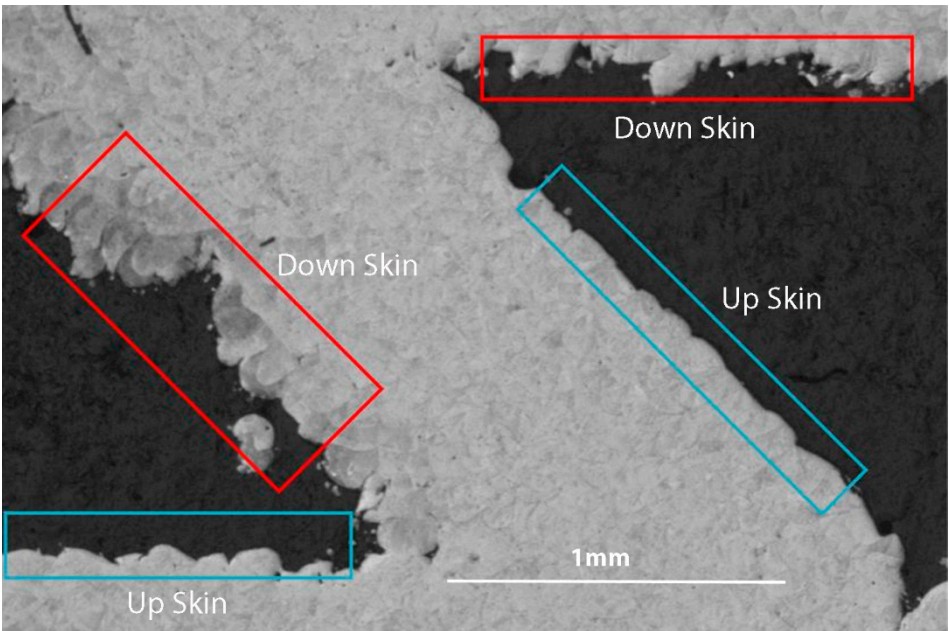

**Figure 13.** Dross formation on down skin surfaces of lattices.

*3.2. Compressive Behaviour*

3.2.1. Elastic Modulus and Yield Strength

The specific elastic moduli and yield strengths (Figure 14) were determined from the stress–strain data for compression in the x (Figure 15a,b) and y directions (Figure 15c,d). Orientations for compression testing are given in Figure 6. The yield strength was determined using the 0.2% offset method. During compression in the y direction, all the samples except sample 3 exhibited sudden collapse by shearing of the layer either near top or bottom of the sample. Sample 3 did not reach its yield point within the load limit of the machine (100kN), and, therefore, no yield strength data point is provided for sample 3 in Figure 14.

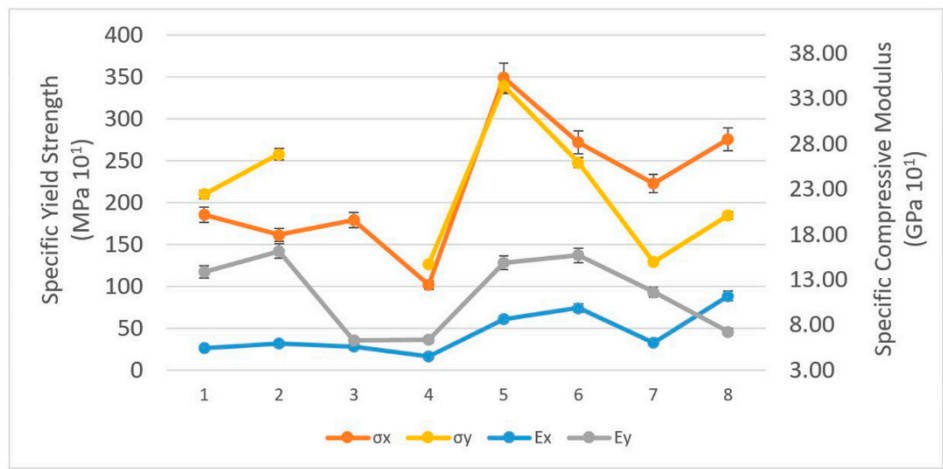

**Figure 14.** Elastic modulus and yield strength for auxetic structures under uniaxial compression.

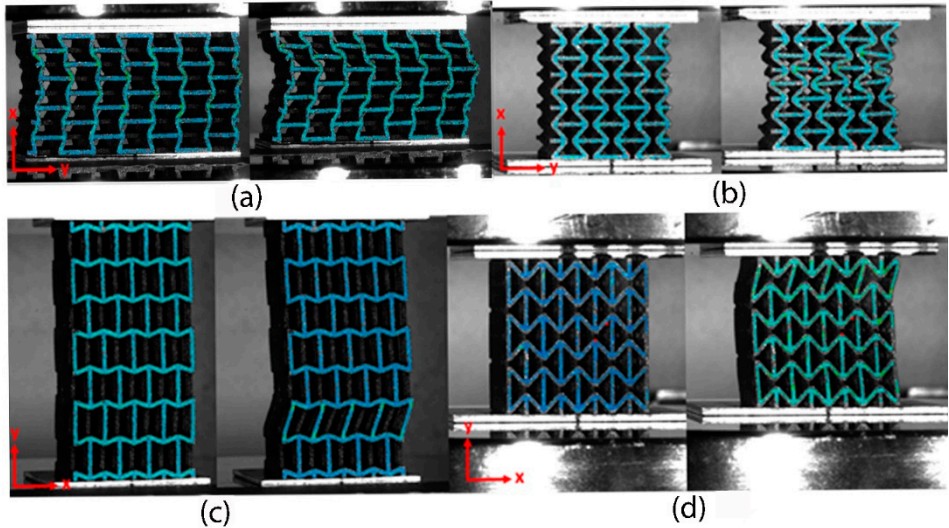

**Figure 15.** DIC images. (**a**) and (**b**): deformation of samples under compression in the x direction; (**c**) and (**d**): deformation of samples under compression in the y direction.

There is minimal variation in Ex (0.89 ± 0.02 GPa–1.91 ± 0.05 GPa, a range of 1.02 ± 0.07 GPa) across the different auxetic structures, whereas Ey varies considerably (from 0.72 ± 0.02 GPa–5.2 ± 0.1 GPa, a range of 4.4 ± 0.1 GPa). There is considerable variation in both σx and σy, each varying similarly with structure type.

Buckling of the auxetic structures is observed in Figure 15, and this is the common mode of collapse, and failure, of an auxetic structure [43], who were able to provide a model the buckling mode for three cell edges for conventional hexagonal honeycombs of uniform thickness edges under uniaxial compression, concluding that buckling became more likely as density was reduced. Their model was extended by Wang and McDowell to other cell geometries, including square, triangular, Kagome and diamond [44]. Recent attempts have been made to make auxetic structures more resilient to buckling collapse, using reinforced structures such as the rhombic reinforced normal re-entrant hexagonal honeycomb (NRHH), which displayed improved buckling strength over the unreinforced NRHH structure [45].

### 3.2.2. Poisson's Ratio

The longitudinal and transverse strain were calculated by using the virtual extensometer as illustrated in Figure 16. Poisson's ratio of the whole structure may not be the same as the unit cell in

the middle of the structure due to presence of boundary effects through compression in both the x and y directions. The boundary effects during compression in the y direction are due to an imbalance of force and moment created due to the termination of structure, which causes the vertical struts to deform more. Similar boundary effects were observed in the work carried out by Yang [46], who also stated that the deformation of the vertical strut due to boundary effect increases with the increase in the length of the vertical strut. Therefore, it was necessary to analyse Poisson's ratio at various regions of the structure in order to determine the degree to which Poisson's ratio of the unit cell at the centre represent the whole structure. Measurement of transverse and longitudinal strain was made in four regions of structure 6, transverse strain being measured in top, middle and bottom layers in regions A and B. Longitudinal strain was measured in the left, middle and right layers of region A and in the left and right layers in region B, as illustrated in Figure 17a,b (x and y direction, respectively). From the results (Table 3), it can be seen that ν of the unit cell in the centre (C and D) is higher than other regions. The error in ν in Table 3 is largely due to only single samples being taken. Furthermore, ν decreases as it proceeds towards outside, which shows that the ν of the unit cell in the centre cannot represent the whole re-entrant structure under compression in the x direction. The influence of end effects in ν can be reduced by fabricating large samples by increasing the number of unit cells. The calculated Poisson's ratios, using the averaging method, for all samples, are given in Figure 18.

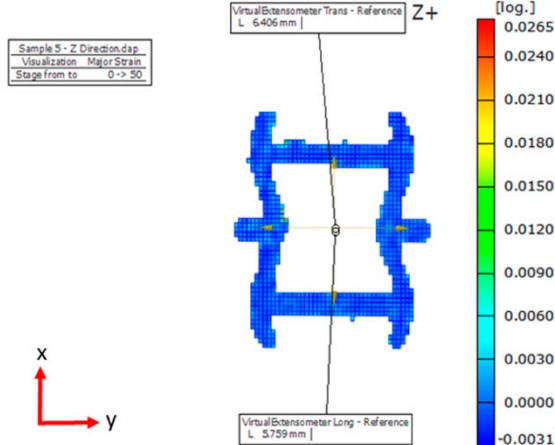

**Figure 16.** Illustration of the measurement of transverse and longitudinal strain from the DIC image.

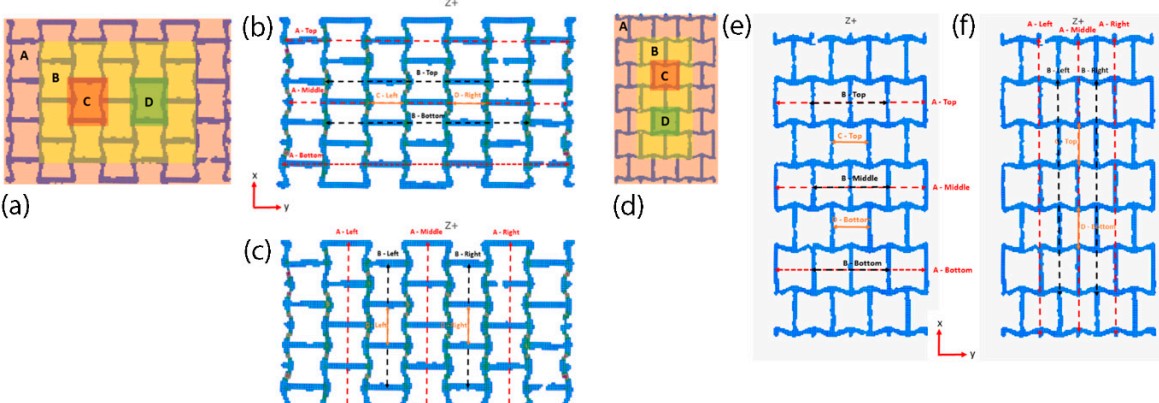

**Figure 17.** Poisson's ratio DIC strain analysis. (**a**) Location of unit cells for compression in x, (**d**) location of unit cells for compression in y, (**b**) measurement of transverse strain in the x direction, (**c**) measurement of longitudinal strain in the x direction, (**e**) measurement of transverse strain in the y direction, (**f**) measurement of longitudinal strain in the y direction.

**Table 3.** Poisson's ratio values for different unit cell locations in sample 6.

| Region | Poisson's Ratio (ν) Compression in x Direction | Poisson's Ratio (ν) Compression in y Direction |
|---|---|---|
| A | −0.6 ± 0.5 | 0.0 ± 0.1 |
| B | −0.7 ± 0.6 | −0.1 ± 0.1 |
| C | −1.2 ± 0.8 | −0.2 ± 0.1 |
| D | −1.3 ± 0.7 | −0.2 ± 0.1 |
| Average | −0.9 ± 0.7 | −0.1 ± 0.1 |
| SD | 0.34 | 0.09 |
| Error | 18% | 43% |

It can be seen that there is little variation in Poisson's ratio for compression in the y direction, but large deviation in values is observed for compression in the x direction. Although samples 2 and 3 have similar relative density, there is a significant variation in E and σ in both directions and of Poisson's ratio in the x direction. Similar variation can also be seen for samples 6 and 8 with similar relative density. This indicates that there is no correlation between relative density and Poisson's ratio. Yang stated that Poisson's ratio would contribute to the difference in E and σ [46], but as observed here, this is true only for compression in x. The number of unit cells in the x and y directions were equal for all samples, and, thus, the overall dimensions of the samples differed, However, comparing samples 3 and 7 in Figure 5, there does not appear to be any correlation between variation in overall size and Poisson's ratio for compression in x.

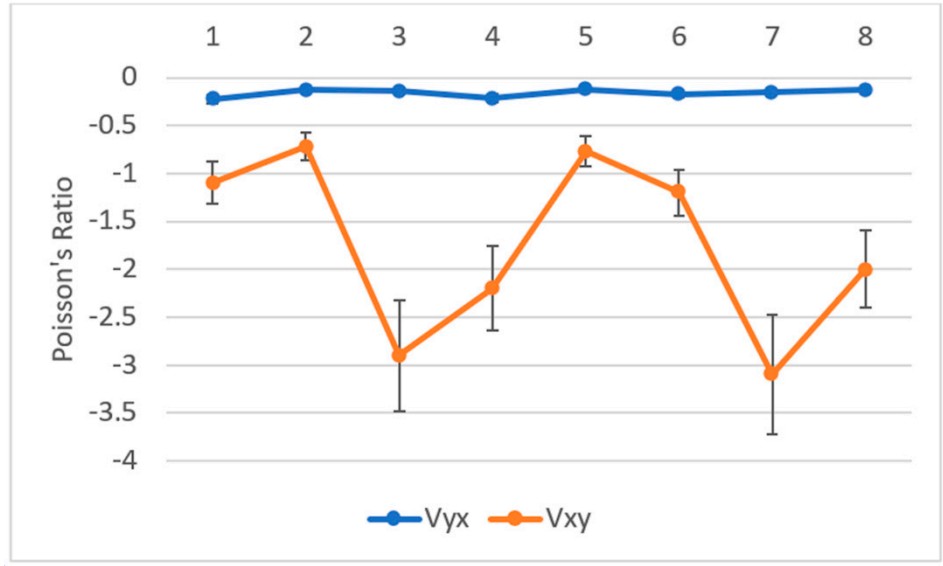

**Figure 18.** Poisson's ratio for compression of auxetic structures along x and y axes.

### 3.3. ANOVA Investigation of Parameter Effects of E, σ and ν.

To understand the influence of unit cell parameters on E, σ and ν, the results were analysed using analysis of variance (ANOVA), using Minitab 19 (Minitab LLC, Chicago, USA). Due to the half factorial DOE model, the analysis was restricted to study only the main effects of the unit cell parameters without interaction. Level of significance (α) of 0.05 was chosen for this study. The strength of the main effects (F-factors) and significance level (P-Values) are given in Table 4 ($E_x$ and $E_y$), Table 5 ($\sigma_x$ and $\sigma_y$) and Table 6 ($\nu_{xy}$ and $\nu_{yx}$). The magnitude of the effect of each parameter is given in Table 7.

**Table 4.** ANOVA output for $E_x$ and $E_y$.

| Parameter | Young's Modulus ($E_x$) x Direction | | Young's Modulus ($E_y$) y-Direction | |
|---|---|---|---|---|
| | F-Value | *p*-Value | F-Value | *p*-Value |
| H | 0.3 | 0.623 | 0.26 | 0.644 |
| L | 3.36 | 0.164 | 35.85 | 0.009 |
| θ | 2 | 0.252 | 7.92 | 0.067 |
| t | 20.92 | 0.02 | 6.31 | 0.087 |

Grey box indicates statistical significance. $R^2$ = 89.86% ($E_x$), 94.38% ($E_y$)

**Table 5.** ANOVA output for $\sigma_x$ and $\sigma_y$.

| Parameter | Yield Strength ($\sigma_x$) x Direction | | Yield Strength ($\sigma_y$) y-Direction | |
|---|---|---|---|---|
| | F-Value | *p*-Value | F-Value | *p*-Value |
| H | 9.7 | 0.053 | 0.1 | 0.779 |
| L | 115.88 | 0.002 | 31.23 | 0.031 |
| θ | 0.54 | 0.515 | 10.64 | 0.083 |
| t | 187.32 | 0.001 | 22.1 | 0.042 |

Grey box indicates statistical significance. $R^2$ = 99.05% ($\sigma_x$), 97.68% ($\sigma_y$)

**Table 6.** ANOVA output for $\nu_{xy}$ and $\nu_{yx}$.

| Parameter | Poisson's Ratio ($\nu_{xy}$) x Direction | | Poisson's Ratio ($\nu_{yx}$) y Direction | |
|---|---|---|---|---|
| | F-Value | *p*-Value | F-Value | *p*-Value |
| H | 2.15 | 0.239 | 0.02 | 0.895 |
| L | 32.39 | 0.011 | 0.02 | 0.895 |
| θ | 0.05 | 0.837 | 4.59 | 0.122 |
| t | 1.07 | 0.377 | 12.76 | 0.038 |

Grey box indicates statistical significance. $R^2$ = 92.24% ($\nu_{xy}$), 85.29% ($\nu_{yx}$)

**Table 7.** Parameter effects for each measurable.

| Parameter | Effect | | | | | |
|---|---|---|---|---|---|---|
| | $E_x$ (GPa) | $E_y$ (GPa) | $\sigma_x$ (MPa) | $\sigma_y$ (MPa) | $\nu_{xy}$ | $\nu_{yx}$ |
| H | −0.064 | +0.195 | −6.44 | −1.65 | +0.408 | −0.003 |
| L | −0.215 | −2.274 | −22.27 | −28.71 | −1.583 | +0.003 |
| θ | +0.166 | −1.069 | −1.52 | −16.76 | −0.063 | +0.038 |
| t | +0.536 | +0.954 | +28.32 | +24.15 | −0.288 | +0.063 |

Grey box indicates statistical significance.

There is only one significant parameter controlling E, that is, "strut thickness (t)" for $E_x$ and "re-entrant strut length (L)" for $E_y$. From Table 7, t is seen to have a positive correlation with $E_x$, increasing stiffness by an average of 0.536 GPa when increasing nominal strut thickness from 0.8 to 1.0 mm. Re-entrant strut length (L), on the other hand, has a significant negative effect on $E_y$, reducing stiffness by 2.274 GPa when moving from an L of 2.5 to 3.5 mm. For σ, there are two significant controlling parameters, t and L, with t being slightly more dominant for $\sigma_x$ and L having the larger effect for $\sigma_y$. For both $\sigma_x$ and $\sigma_y$, increasing L from 2.5 mm to 3.5 mm has a negative correlation with σ, reducing it by 22.27 MPa ($\sigma_x$) and 28.71 MPa ($\sigma_y$). Conversely, t has a positive correlation with both $\sigma_x$ and $\sigma_y$, increasing them by 28.32 and 24.15 MPa, respectively. For Poisson's ratio (ν), there is one factor only that is dominant, although for $\nu_{xy}$, it is L, and for $\nu_{yx}$, it is t. L has a large negative correlation with $\nu_{xy}$, reducing it by 1.583 when moving from 2.5 to 3.5 mm in length. t has a small positive correlation

with $\nu_{yx}$, increasing it by 0.063 when moving from 0.8 mm to 1.0 mm in strut thickness. Neither H (vertical strut length) nor θ (re-entrant strut angle) have a significant effect on any of the measured mechanical properties. The root causes of these observations may be elucidated by understanding the deformation mechanisms dominant in the structures, as discussed in Section 3.4.

### 3.4. Deformation Mechanism

From the literature review, it was found that the re-entrant structure deforms by hinging, stretching and bending of re-entrant struts. However, many researchers considered theoretically that bending had been the dominant mechanism. Yang, however, stated that the axial compression of vertical struts also plays a major role in the deformation of the re-entrant structure when compressed in the y direction [46]. As there is lack of clarity in the deformation mechanism, there is a need to study the structures under compression using the digital correlation technique (DIC). DIC data from sample 6 were used for studying the elastic deformation of the structures under compression.

### 3.4.1. Compression in the x Direction

The deformation mechanism of re-entrant structure under compression in the x direction was analysed through measurement of the change in length of the struts and re-entrant strut angle under compression (Figure 19a). Strains measured for re-entrant struts L1 and L2 indicate compression (negative strain), and strain measured for H indicates tension (positive strain). Strains measured for A1, A2, B1 and B2 (Figure 19b) indicate that A1 and A2 undergo compression and B1 and B2 undergo tension (Table 8), resulting in a bending of the strut, as illustrated in Figure 19c. There is also an increase in the re-entrant angle, θ. The results indicate that the unit cell deformation through compression in the x direction thus occurs as a result of a combination of vertical strut stretching, re-entrant strut compression, bending and hinging.

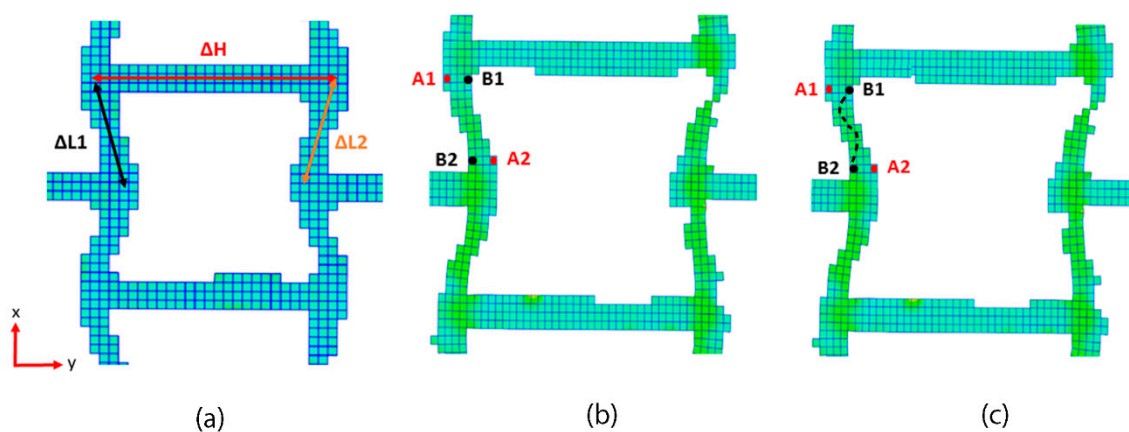

|     | (a) | (b) | (c) |

**Figure 19.** (**a**) Struts measured for dimensional change under compression in the x direction, (**b**) points analysed to assess degree of strut bending, (**c**) illustration of strut bending.

**Table 8.** Calculated strut length changes and re-entrant angle change for compression in the x direction.

|                  | ΔL1 (mm) | ΔL1 (mm) | ΔH (mm)  | Δθ (°)   |
|------------------|----------|----------|----------|----------|
| Dimension Change | −0.06    | −0.06    | +0.08    | 4.77     |
|                  | A1 (mm)  | A2 (mm)  | B1 (mm)  | B2 (mm)  |
| Dimension Change | −0.004   | −0.001   | +0.025   | +0.046   |

### 3.4.2. Compression in the y Direction

A similar analysis of the auxetic structure was made for compression in the y direction using DIC. The dimension and points used for analysis are given in Figure 20. Strains measured for re-entrant struts

L1 and L2 indicate tension (positive strain), and strain measured for H indicates axial compression (negative strain). Strains measured for A1, A2, B1 and B2 (Figure 20b) indicate that A1 and A2 undergo tension and B1 and B2 undergo compression (Table 9), resulting in a bending of the strut, as illustrated in Figure 20c. There is also an increase in the re-entrant angle, θ, although this is not as high as that observed for compression in the x direction (Table 8), and the hinging mechanism is, therefore, less dominant in the y direction compression-induced unit cell deformation than it was in the x direction compression-induced unit cell deformation. The results indicate that the unit cell deformation through compression in the y direction thus results predominantly from a combination of vertical strut compression and re-entrant strut stretching and bending.

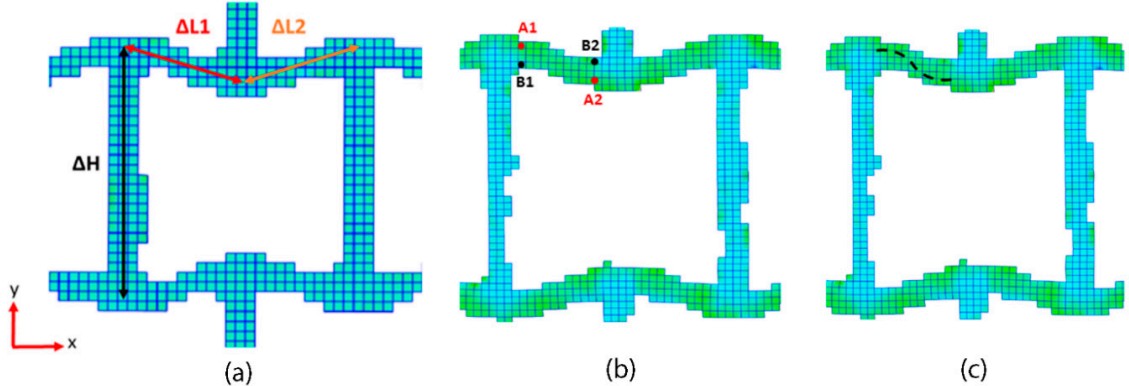

**Figure 20.** (**a**) Struts measured for dimensional change under compression in the y direction, (**b**) points analysed to assess degree of strut bending, (**c**) illustration of strut bending.

**Table 9.** Calculated strut length changes and re-entrant angle change for compression in the y direction.

|  | ΔL1 (mm) | ΔL1 (mm) | ΔH (mm) | Δθ (°) |
|---|---|---|---|---|
| Dimension Change | −0.06 | −0.06 | +0.08 | 4.77 |
|  | A1 (mm) | A2 (mm) | B1 (mm) | B2 (mm) |
| Dimension Change | −0.004 | −0.001 | +0.025 | +0.046 |

## 4. Conclusions

This paper has presented a manufacturability study on 3D re-entrant structures using PBF and analysis of their mechanical properties. Twenty-four half-factorial design of experiments (DOE) were designed to study the influence of unit cell parameters, namely, the length of the vertical strut 'H', the length of the re-entrant strut 'L', the re-entrant angle 'θ' and the strut thickness 't' on the mechanical properties of the structure.

It has been demonstrated that auxetic 3D structures can be successfully built using laser powder bed fusion (PBF) with a density > 99.3% for all geometries.

Porosity in the struts is found to be dependent on the orientation of the strut to the vertical z-axis. Lowest porosity is achieved for struts aligned nearest to the z-axis, and highest porosity was found on struts aligned furthest away from the z-axis. Porosity levels are higher nearer the strut surfaces and the become less circular.

The geometry of the structure has a significant effect on the accuracy to which it can be manufactured. Struts with alignment nearer to the vertical (Z) axis have the best accuracy (<15 μm thickness deviation from nominal). Those built at low angle to the vertical exhibit the largest deviation from nominal (>300 μm thicker than nominal). The strut length accuracy depends on type of strut within the structure. Vertical struts were built with greatest accuracy (<2.5% deviation from nominal), and re-entrant struts showed the greatest levels of length inaccuracy (7–18%). The level of inaccuracy was lowest for struts built aligned nearer to the Z-axis. The re-entrant angle was sensitive to geometry

(and thus to alignment to the Z-axis), with an average of 5% deviation across all structures, but was 4x higher on average for the samples with 45° re-entrant angles than those with 70° re-entrant angles.

There is minimal variation (1.02 ± 0.07 GPa) in the elastic modulus ($E_x$) measured in the x direction (perpendicular to the horizontal strut) across all auxetic structures, whereas there was considerable variation (4.4 ± 0.1 GPa) observed for $E_y$ (measured parallel to the vertical strut).

There is considerable variation in the yield stresses measured in both x and y directions ($\sigma_x$ and $\sigma_y$), each varying similarly with structure type. Averages for yield stress are within errors equal for both test directions ($\overline{\sigma}_x = 44 \pm 6$, $\overline{\sigma}_y = 52 \pm 10$).

Poisson's ratio for compression in the x direction, $\nu_{xy}$, was found to not vary considerably with geometry type, whereas for compression in the y direction, $\nu_y$ was found to vary considerably with geometry. Poisson's ratio of the unit cell at the centre cannot represent the whole structure due to the presence of end effects. $\nu_{xy}$ decreases as it proceeds to the outside. However, the influence of the end effects on $\nu_{xy}$ can be reduced by fabricating samples with large arrays of unit cells.

There was no correlation found between relative density and Poisson's ratio.

The statistical analysis of mechanical behaviour showed that vertical strut height (H) and re-entrant strut angle ($\theta$) do not have any statistical influence on the mechanical performance of the auxetic structures. $E_x$ is only controlled by strut thickness (t), and $E_y$ is only controlled by re-entrant strut length (L). Both $\sigma_x$ and $\sigma_y$ are controlled by t and L to a similar extent. $\nu_{xy}$ is strongly determined by L, and $\nu_{yx}$ is controlled weakly by t.

The results indicate that the mechanisms of unit cell deformation are different for compression in x and y. For compression in the x direction, deformation occurs as a result of a combination of vertical strut stretching, re-entrant strut compression, bending and hinging, whereas in the y direction, deformation results predominantly from a combination of vertical strut compression and re-entrant strut stretching and bending.

This research has demonstrated that control over the mechanical properties of 3D auxetic structures manufactured using additive manufacturing (AM) is achievable through the design of the unit cells and primarily through control of the strut thickness and re-entrant strut length. AM offers the ability to fabricate these structures with controlled design and controlled mechanical response, and, as such, although not the focus of this research, their energy impact properties could also be tailored, enabling their extended application in space, defence and auto applications as crash/impact protection structures. Further research directions are to be aimed at addressing the affordability of manufacturing such structures through AM, which is a high-value manufacturing technology. We aim to investigate the emerging higher rate AM technologies that can offer economies of scale, but require validation for auxetic structure validation.

**Author Contributions:** Conceptualisation, G.J.G. and I.G.M.; methodology, S.M.; formal analysis, S.M.; investigation, S.M.; resources, S.M.; data curation, G.J.G.; writing—original draft preparation, G.J.G.; writing—review and editing, G.J.G., S.M.; supervision, G.J.G., I.G.M.; project administration, G.J.G. All authors have read and agreed to the published version of the manuscript.

**Funding:** This research received no external funding.

**Acknowledgments:** The authors wish to thank WMG for supporting the MSc, the results of which form the basis of this article.

**Conflicts of Interest:** The authors declare no conflict of interest.

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
