# Peer review of "Additive Manufacture of 3D Auxetic Structures by Laser Powder Bed Fusion—Design Influence on Manufacturing Accuracy and Mechanical Properties"

_applsci, doi:10.3390/app10217738_

Round 1

Reviewer 1 Report

The article details the laser beam manufacturing and the mechanical behavior of a specific type of auxetic structure, with a DOE approach to evaluate the influence of geometric factors on the elastic properties. The article is especially interesting for its discussion on the minimization of defects such as porosities and "dross" by adjusting processing parameters. The use of DIC to follow the deformation of these auxetic structures is very interesting too.

The reviewer has one major question concerning this study. It seems problematic to me that the dimensions of the specimen are not always the same depending on the geometric factors. I understand this is done to preserve the same number of unit cells in the three dimensions. However if one consider this metallic foam as a homogeneous equivalent material, then doing compression on parallepipeds of different dimensions is questionable. By having different ratios of height/surface, the sensitivity to buckling and to flow localization (such as seen on figures 15a and c) is modified. To me it becomes difficult to know if the local deformation of the second line of cells (from the bottom) on figure 15c is coming from the architecture, or if it is triggered by the fact that this specimen is higher than others. Moreover in compression, friction is not negligible (even if here the material is a foam), and having different areas of contact with the anvils for the different specimen is questionable. Therefore I was wondering if this had any effect on the mechanical properties. For future studies, using cubic Representative Elementary Volumes (REV) would avoid these questions, because the macroscopic geometry would always be the same for all the compression tests.

The reviewer advises to plot the elastic modulus of figure 14 as a function of density, even if the correlation with density is not obvious. Otherwise the reader cannot estimate the "specific" properties. Also it would give a graphical estimation of a possible agreement with Gibson-Ashby model, or Voigt-Reuss bounds. Plotting these bounds on the graph would help a lot too.

For future works, using FEM modeling together with DIC would be very interesting. Also having other mechanical tests than compression would be of interest : if shear test is possible, then it would be possible to identify all the components of the stiffness tensor, which would be extremely useful for the reader.

The paper is of good quality and could be published if and only if the authors can proove that there is no bias coming from the different geometries of the REV of their specimen.

Author Response

The authors wish to thank the reviewer for their constructive and useful review. With regards to their comment regarding the effect of geometry size on the mechanical properties, the authors accept that there may be some effect due to the difference in dimensions of the samples, but as the reviewer appreciates, the authors wished to maintain equal and whole numbers of unit cells in each sample, which resulted in differences in sample volumes. The authors believe that as ANOVA was performed, any disturbance in the observed significance of the main parameters would be observed as an increase in the size of the noise signal in the ANOVA response. The values for the error response are not excessive. The R2 values have been included for each response in Tables 4-7. 

Figure 14 has been plotted as a function of density.

In further work we will be assessing other mechanical properties of the structures, including shear.

Reviewer 2 Report

The paper is well wiritten and presents clearly the work of the authors regarding AM manufactured auxetic structures.

I just have one small suggestion, which may help to widen the scope of the conducted work. It would be beneficial to describe future research directions and/or which applications may be addressed using such structures in real components.

Author Response

The authors wish to thank the reviewer for his time in providing this review. In response to their comment:

'It would be beneficial to describe future research directions and/or which applications may be addressed using such structures in real components.'

we have included the following at the end of the conclusions:

'This research has demonstrated that control over the mechanical properties of 3D auxetic structures manufactured using Additive Manufacturing (AM) is achievable through the design of the unit cells, and primarily through control of the strut thickness, re-entrant strut length. AM offers the ability to fabricate these structures with controlled design and controlled mechanical response, and as such, although not the focus of this research, their energy impact properties could also be tailored, enabling their extended application in space, defence and auto applications as crash/impact protection structures. Further research directions are to be aimed at addressing the affordability of manufacturing such structures through AM, which is a high-value manufacturing technology. We aim to investigate the emerging higher rate AM technologies that can offer economies of scale, but require validation for auxetic structure validation.'

Reviewer 3 Report

Report on manuscript ”Additive Manufacture of 3D Auxetic Structures by Laser Powder Bed Fusion – Design Influence on Manufacturing Accuracy and Mechanical Properties” Sibi Maran, Iain G. Masters and Gregory J. Gibbons

In the manuscript, the authors present both the manufacturing process for 3D re-entrant structures using Laser Powder Bed Fusion and the influence of unit cell parameters on the mechanical properties of obtained samples. The study is important for engineers, especially those who develop products using auxetic materials and structures, i.e. systems with negative Poisson’s ratio. Although the studied structure is well known in the literature and has been widely investigated experimentally, theoretically, and by computer simulations, the article can be thought of as original due to the applied materials - metal. Moreover, the study is topical because of rapidly growing interest in application of auxetics. Hence, in my opinion, the paper is worth publishing. However, prior to acceptance, the authors should consider and respond to the remarks below.

It would be good if the authors could emphasize the rapidly growing interest in auxetics in the introduction and enrich the introduction by discussing relevant works on auxetics.

The works mentioned below seem to be close to the subject of this article:
1) Effect of Two-Dimensional Re-Entrant Honeycomb Configuration on Elastoplastic Performance of Perforated Steel Plate By: Y.Y.Zhou, C.F. Zhou, Z. Shu, L.J. Jia, APPLIED SCIENCES-BASEL, Volume: 10, Article Number: 3067, Published: MAY 2020,
2) Smart Honeycomb "Mechanical Metamaterials" with Tunable Poisson's Ratios”, By: J.N. Grima-Cornish, R. Cauchi, D. Attard, R. Gatt, J.N. Grima, PHYSICA STATUS SOLIDI B-BASIC SOLID STATE PHYSICS, Article Number: 1900707, DOI: 10.1002/pssb.201900707, Early Access: JUL 2020.

There are mistakes and inaccuracies in references. For example, in line 41 Larsen et al [7] and in references in line 446, one can find the following text: “7. Larsen, U.; Sigmund, O. Foam structures with a negative Poisson’s ratio. Science 1987, 235, pp. 1038-1040” that is not correct because “Foam structures with a negative Poisson’s ratio. Science 1987, 235, pp. 1038-1040” is the information about Lakes’ paper but not Larsen. Further, in line 449 instead “Grima, C.” should be “Grima J.” Next, line 72 the name of the author in the text “Hussain” is different than in the References (line 468) “Hussein”. All references should be carefully checked.

Regarding the information given in lines 41-43: “the Chiral Honeycomb model, first introduced by Prall and Lakes [8], made up of circular nodes connected by the straight ligaments in a tangential manner; “ – the model discussed in ref. [8] has been presented by the 2nd author already at the beginning of 90s. Moreover, it seems that the authors are not aware that much earlier (in 80s) chiral molecular models of hexamer molecules, exhibiting isotropic phase of negative Poisson’s ration. have been presented by Wojciechowski in the below papers:
a) describing computer simulations of two dimensional model molecules (hard cyclic hexamers) forming isotropic phase which exhibits negative Poisson’s ratio: [“CONSTANT THERMODYNAMIC TENSION MONTE-CARLO STUDIES OF ELASTIC PROPERTIES OF A TWO-DIMENSIONAL SYSTEM OF HARD CYCLIC HEXAMERS”; By: K. W. WOJCIECHOWSKI, MOLECULAR PHYSICS, Volume: 61, Pages: 1247-1258, Published: AUG 10 1987],
b) presenting the first rigorously solved model (soft cyclic hexamers) with a stable isotropic (chiral) phase of negative Poisson’s ratio (auxetic phase) was: [“TWO-DIMENSIONAL ISOTROPIC SYSTEM WITH A NEGATIVE POISSON RATIO”; By: K. W. WOJCIECHOWSKI, PHYSICS LETTERS A, Volume: 137, Pages: 60-64, Published: MAY 1 1989].

More recently a strongly anisotropic chiral phase of negative Poisson’s ratio was found in a work about hard cyclic tetramers [“Auxetic, Partially Auxetic, and Nonauxetic Behaviour in 2D Crystals of Hard Cyclic Tetramers”; by Konstantin V. Tretiakov and Krzysztof W. Wojciechowski, Physica Status Solidi RRL, Volume 14, Article Number: 2000198, Published: 2020].

The mechanism of the rotating squares [9] has been described independently by Ishibashi and Iwata in [Y. Ishibashi and M. Iwata, “A microscopic model of a negative Poisson's ratio in some crystals”, JOURNAL OF THE PHYSICAL SOCIETY OF JAPAN, Volume: 69, Pages: 2703-2702, Published: AUG 2000].

Line 74: “CT” is not defined in the text.

A scale bar in Figure 13 would be useful.

In Figure 15, one can observe that buckling occurs in some structures. Buckling has been recently intensively studied in the context of auxetics. Might the authors comment that in their paper?

It seems that the (very large) statistical errors presented in table 3 concern single measurements instead of the average results; in the latter case the error should be divided by the square root of the number of ‘single’ measurements. Might the authors comment this in their paper?

Was there any size dependence of the obtained results analyzed? The measured elastic properties, and Poisson’s ratio in particular, may show essential dependence on the number of (repeating) cells used and the boundary conditions applied. Might the authors comment this in their paper?

Is the dashed line shown in Figure 19c plotted properly? Looking at that figure, one would expect the opposite convexity/concavity. Might the authors comment this in their paper?

In the Conclusions one can read “There was no correlation found between relative density and mechanical properties.” Do the authors suggest that this concerns also the Young’s modulus?

Author Response

The authors wish to thank the reviewer for their considerable time spent on providing a detailed set of comments. We have agreed with all the reviewer's criticisms and have responded as below:

The works mentioned below seem to be close to the subject of this article:
1) Effect of Two-Dimensional Re-Entrant Honeycomb Configuration on Elastoplastic Performance of Perforated Steel Plate By: Y.Y.Zhou, C.F. Zhou, Z. Shu, L.J. Jia, APPLIED SCIENCES-BASEL, Volume: 10, Article Number: 3067, Published: MAY 2020,
2) Smart Honeycomb "Mechanical Metamaterials" with Tunable Poisson's Ratios”, By: J.N. Grima-Cornish, R. Cauchi, D. Attard, R. Gatt, J.N. Grima, PHYSICA STATUS SOLIDI B-BASIC SOLID STATE PHYSICS, Article Number: 1900707, DOI: 10.1002/pssb.201900707, Early Access: JUL 2020.

A section has been included in the literature review to highlight some of the recent interest in auxetic structures and 4 references added related to their application in civil engineering and smart metamaterials - ‘There has been a recent rapid growth in interest in auxetic structures, particularly in their application in civil engineering, e.g as damping elements for seismic protection [21], and as damage tolerant building materials [22]; and as new smart metamaterials [23], [24].’

There are mistakes and inaccuracies in references. For example, in line 41 Larsen et al [7] and in references in line 446, one can find the following text: “7. Larsen, U.; Sigmund, O. Foam structures with a negative Poisson’s ratio. Science 1987, 235, pp. 1038-1040” that is not correct because “Foam structures with a negative Poisson’s ratio. Science 1987, 235, pp. 1038-1040” is the information about Lakes’ paper but not Larsen.

This has been corrected to ‘Larsen, U.D.; Sigmund, O.; & Bouwstra, S. Design and Fabrication of Compliant Mecromechanisms and Structures with Negative Poisson’s Ratio. J. Microelectromech. Syst. 1997, 6(2), pp. 365–371

Further, in line 449 instead “Grima, C.” should be “Grima J.” Corrected.

Next, line 72 the name of the author in the text “Hussain” is different than in the References (line 468) “Hussein”.

Name in body of text corrected to ‘Hussein [27], manufactured four..’

All references should be carefully checked. Checked.

Regarding the information given in lines 41-43: “the Chiral Honeycomb model, first introduced by Prall and Lakes [8], made up of circular nodes connected by the straight ligaments in a tangential manner; “ – the model discussed in ref. [8] has been presented by the 2nd author already at the beginning of 90s.

This has been amended. A new article has been included ‘R. S. Lakes, Deformation mechanisms of negative Poisson's ratio materials: structural aspects. J. Mater. Sci 1991, 26, pp. 2287-2292’ and the text modified ‘the Chiral Honeycomb model, first introduced by Lakes [10], and further described by Prall and Lakes [11]’

Moreover, it seems that the authors are not aware that much earlier (in 80s) chiral molecular models of hexamer molecules, exhibiting isotropic phase of negative Poisson’s ration. have been presented by Wojciechowski in the below papers:
a) describing computer simulations of two dimensional model molecules (hard cyclic hexamers) forming isotropic phase which exhibits negative Poisson’s ratio: [“CONSTANT THERMODYNAMIC TENSION MONTE-CARLO STUDIES OF ELASTIC PROPERTIES OF A TWO-DIMENSIONAL SYSTEM OF HARD CYCLIC HEXAMERS”; By: K. W. WOJCIECHOWSKI, MOLECULAR PHYSICS, Volume: 61, Pages: 1247-1258, Published: AUG 10 1987],
b) presenting the first rigorously solved model (soft cyclic hexamers) with a stable isotropic (chiral) phase of negative Poisson’s ratio (auxetic phase) was: [“TWO-DIMENSIONAL ISOTROPIC SYSTEM WITH A NEGATIVE POISSON RATIO”; By: K. W. WOJCIECHOWSKI, PHYSICS LETTERS A, Volume: 137, Pages: 60-64, Published: MAY 1 1989].

The introduction has been modified to include this work ‘Wojciechowski developed computer models of hard [5] and soft [6] cyclic hexamers and demonstrated that they exhibited phases having negative Poisson’s ratio.’

The two new references [5] and [6] have been added to the bibliography.

More recently a strongly anisotropic chiral phase of negative Poisson’s ratio was found in a work about hard cyclic tetramers [“Auxetic, Partially Auxetic, and Nonauxetic Behaviour in 2D Crystals of Hard Cyclic Tetramers”; by Konstantin V. Tretiakov and Krzysztof W. Wojciechowski, Physica Status Solidi RRL, Volume 14, Article Number: 2000198, Published: 2020].

This has been appreciated and the text in the introduction has been modified ‘Wojciechowski developed computer models of hard [5] and soft [6] cyclic hexamers and demonstrated that they exhibited phases having negative Poisson’s ratio, and more recently, a strongly anisotropic chiral phase of negative Poisson’s ratio was found in a work about hard cyclic tetramers [7].’

A new reference [7] has been added to the bibliography.

The mechanism of the rotating squares [9] has been described independently by Ishibashi and Iwata in [Y. Ishibashi and M. Iwata, “A microscopic model of a negative Poisson's ratio in some crystals”, JOURNAL OF THE PHYSICAL SOCIETY OF JAPAN, Volume: 69, Pages: 2703-2702, Published: AUG 2000].

This has been appreciated and the text has been changed in the introduction to ‘.., which was first introduced by Grima and Evans [13] and, independently, by Ishibashi and Iwata [14] in the same year, in the form of rigid squares connected at the vertices by hinges..’

A new reference [14] has been added to the bibliography.

Line 74: “CT” is not defined in the text.

Text changed to ‘Computer Tomography (CT) analysis carried..’

A scale bar in Figure 13 would be useful.

Scale bar added to Figure 13.

In Figure 15, one can observe that buckling occurs in some structures. Buckling has been recently intensively studied in the context of auxetics. Might the authors comment that in their paper?

The authors appreciate this advice and have added text into 3.2.1. “Buckling of the auxetic structures is observed in Figure 15, and this is the common mode of collapse, and failure, of an auxetic structure [44], who were able to provide a model of the buckling mode for three cell edges for conventional hexagonal honeycombs of uniform thickness edges under uniaxial compression, concluding that buckling became more likely as density was reduced. Their model was extended by Wang and McDowell to other cell geometries, including square, triangular, Kagome and diamond [45]. Recent attempts have been made to make auxetic structures more resilient to buckling collapse, using reinforced structures such as the rhombic reinforced normal re-entrant hexagonal honeycomb (NRHH), which displayed improved buckling strength over the unreinforced NRHH structure [46].”

Three additional references have been included in the bibliography [44], [45], [46].

It seems that the (very large) statistical errors presented in table 3 concern single measurements instead of the average results; in the latter case the error should be divided by the square root of the number of ‘single’ measurements. Might the authors comment this in their paper?

The author accepts that the error is large due to single samples being measured. They have acknowledged this in the text L296  ‘The error in ν in Table 3 is large due to only single samples being taken.’

Was there any size dependence of the obtained results analyzed? The measured elastic properties, and Poisson’s ratio in particular, may show essential dependence on the number of (repeating) cells used and the boundary conditions applied. Might the authors comment this in their paper?

This comment is appreciated by the author and the following text was added in 3.2.2 ‘The number of unit cells in the x and y directions were equal for all samples, and thus the overall dimensions of the samples differed, although, comparing samples 3 and 7 in Figure 5, there does not appear to be any correlation between variation in overall size and Poisson’s ratio for compression in x.’

Is the dashed line shown in Figure 19c plotted properly? Looking at that figure, one would expect the opposite convexity/concavity. Might the authors comment this in their paper?

The authors agree with the reviewer and this is an error. The Figure 19c has been modified to show the correct bending motion.

In the Conclusions one can read “There was no correlation found between relative density and mechanical properties.” Do the authors suggest that this concerns also the Young’s modulus?

No, this is not correct. The text in the conclusions has been corrected to read ‘There was no correlation found between relative density and Poisson’s ratio.”

Reviewer 4 Report

In the paper “Additive Manufacture of 3D Auxetic Structures by Laser Powder Bed Fusion – Design Influence on Manufacturing Accuracy and Mechanical Properties” Sibi Maran  Iain G Masters and Gregory J Gibbons describe studies of mechanical properties of some structures which are auxetic, i.e. exhibit negative Poisson’s ratio. The structures are re-entrant and anisotropic.

This is a fair paper. It would be stronger if the authors could revise it taking into account the below remarks.

  1. The 1st sentence of the INTRODUCTION is not only false but it is in contradiction with the 2nd sentence.

  1. In the 6th sentence of the introduction the authors claim that re-entrant structure was observed in auxetic foams. This is a delicate problem, however, because at compression vertices do not change ‘convexity’ as suggested in [5] but ribs which form convex cells in common (nonauxetic) foams do bend themselves when converting into auxetic foams of nonconvex cells. This fact can be clearly seen in computer simulations of two dimensional structures of auxetic foams, see, e.g., the paper by Pozniak et al. in Smart Materials and Structures (2013).

  1. In the INTRODUCTION the authors mention a few simple mechanical models of auxetics. It seems that the authors are not aware that there exist also simple thermodynamic models showing that thermodynamically stable auxetic phases/structures can be spontaneously formed by very simple model molecules.

  1. Might the authors explain why, from infinitely many possible values, they selected the particular values of parameters of the studied structures they present in the paper?

  1. Why the errors in Table 3 are so large?

Author Response

The authors wish to thank the reviewer for their detailed and useful comments. We accept all the criticism and have responded as detailed below:

  1. The 1st sentence of the INTRODUCTION is not only false but it is in contradiction with the 2nd sentence.

First sentence removed.

  1. In the 6th sentence of the introduction the authors claim that re-entrant structure was observed in auxetic foams. This is a delicate problem, however, because at compression vertices do not change ‘convexity’ as suggested in [5] but ribs which form convex cells in common (nonauxetic) foams do bend themselves when converting into auxetic foams of nonconvex cells. This fact can be clearly seen in computer simulations of two dimensional structures of auxetic foams, see, e.g., the paper by Pozniak et al. in Smart Materials and Structures (2013).

Agree. The authors have referred back to the original reference by Lakes and have modified the statement to ‘Re-entrant structure was also observed in auxetic foams formed by the volumetric compression of conventional foams, with ribs that permanently protrude inwards forming a re-entrant structure’

  1. In the INTRODUCTION the authors mention a few simple mechanical models of auxetics. It seems that the authors are not aware that there exist also simple thermodynamic models showing that thermodynamically stable auxetic phases/structures can be spontaneously formed by very simple model molecules.

The authors agree that this was an oversight. They have included added a sentence and two new references to describe the effect in hexamers and tetramers. ‘Wojciechowski developed computer models of hard [5] and soft [6] cyclic hexamers and demonstrated that they exhibited phases having negative Poisson’s ratio, and more recently, a strongly anisotropic chiral phase of negative Poisson’s ratio was found in a work about hard cyclic tetramers [7].’

  1. Might the authors explain why, from infinitely many possible values, they selected the particular values of parameters of the studied structures they present in the paper?

The values for Ď´, H and L we selected based on an analysis of designs within a CAD environment (Materialise Magics), and were within the range that allowed the structures to be manufactured without the use of any internal supporting structures (which would make post-processing impossible). We then selected appropriate extremes for each of the two levels for each parameter that enabled ‘buildability’ for all samples (as a complete set was required for the ANOVA analysis, so as to be able to determine the effect of these parameters on build quality and mechanical properties of the structures. Text added in the document L118 ‘The values selected for H, L and Ď´ were based on an initial virtual analysis of the buildability of the structures within the CAD environment. The values selected would, theoretically, enable all samples to be manufactured successfully (without failure and with no use of supporting structures) and thus allowing subsequent ANOVA to be performed.’

  1. Why the errors in Table 3 are so large?

The errors are large as they are for single sample values. I have made this clear in the text by adding L299 ‘The error in ν in Table 3 is large due to only single samples being taken.’